# UM171 induces a homeostatic inflammatory-detoxification response supporting human HSC self-renewal

Jalila Chagraoui[1], Bernhard Lehnertz[1], Simon Girard[1], Jean Francois Spinella[1], Iman Fares[2], Elisa Tomellini[1], Nadine Mayotte[1], Sophie Corneau[1], Tara MacRae[1], Laura Simon[1], Guy Sauvageau[1,3,4]*

**1** Molecular Genetics of Stem Cells Laboratory, Institute for Research in Immunology and Cancer (IRIC), University of Montreal, Montreal, QC, Canada, **2** Department of Molecular, Cell and Developmental Biology, UCLA, LA, United States of America, **3** Division of Hematology, Maisonneuve-Rosemont Hospital, Montreal, QC, Canada, **4** Department of Medicine, Faculty of Medicine, University de Montreal, Montreal, QC, Canada

* guy.sauvageau@umontreal.ca

## Abstract

Elucidation of the molecular cues required to balance adult stem cell self-renewal and differentiation is critical for advancing cellular therapies. Herein, we report that the hematopoietic stem cell (HSC) self-renewal agonist UM171 triggers a balanced pro- and anti-inflammatory/detoxification network that relies on NFKB activation and protein C receptor-dependent ROS detoxification, respectively. We demonstrate that within this network, EPCR serves as a critical protective component as its deletion hypersensitizes primitive hematopoietic cells to pro-inflammatory signals and ROS accumulation resulting in compromised stem cell function. Conversely, abrogation of the pro-inflammatory activity of UM171 through treatment with dexamethasone, cAMP elevating agents or NFkB inhibitors abolishes EPCR upregulation and HSC expansion. Together, these results show that UM171 stimulates ex vivo HSC expansion by establishing a critical balance between key pro- and anti-inflammatory mediators of self-renewal.

## Introduction

Inflammation is a well recognized mediator of hematopoietic stem and progenitor cell (HSPC) activity. Examples include Toll-Like Receptors (TLR), Tumor Necrosis Factor alpha (TNFa, prostaglandins, IL6, interferons, GM-CSF and eicosanoids [1, 2]. Most of these inflammation mediators promote the activation of NFkB and subsequent transcriptional induction of pro-inflammatory genes [3, 4]. Depending on the context, inflammation can either promote HSC expansion or differentiation [5–18]. Defining the proper settings in which inflammation mediates HSC self-renewal remains a challenging question.

In previous studies, we showed that the small molecule UM171 promotes the *ex vivo* expansion of HSCs with long-term multi-lineage reconstitution potential [19] and identified Endothelial Protein C Receptor (EPCR), a previously suggested NFkB inhibitor [20–22], as a

**Data Availability Statement:** The sequencing data presented in this manuscript were uploaded to NCBI (accession codes:GSE57561 and GSE138487

and GSE138680). All other relevant data are within the paper and its Supporting Information files.

**Funding:** This work was supported by grants from the Canadian Institutes of Health Research and the Stem Cell Network of Canada (G.S.). E. Tomellini and L. Simon were supported by a fellowship from the Cole Foundation. J.F.S was supported by IVADO scholarship. The funders had no role in study design, data collection and analysis, decision to publish, or preparation of the manuscript.

**Competing interests:** The authors have declared that no competing interests exist.

reliable marker of expanded HSCs [23]. Interestingly, Cohen *et al.*, have demonstrated that EPCR/PAR1 signaling promotes mouse bone marrow retention, survival and stress resistance by limiting inflammation associated nitric oxide production [24]. Here, we show that UM171 induces a strong NFkB-dependent pro-inflammatory signalling together with an EPCR mediated anti-inflammatory response leading to HSC self-renewal. The essential coordination by UM171 of these 2 distinct and necessary networks in primitive HSPC is described.

# Results

## UM171 triggers a specific inflammatory response in HSPC

In preliminary experiments using hematopoietic cell lines, we found that UM171 triggers simultaneous activation of pro-inflammatory (including IKK/NFkB) and anti-inflammatory (including detoxification responses) programs (Fig 1A–1C and S1 Fig). These programs are activated as early as 6 hours post-treatment, with the most significant changes reached at 48-72h.

To characterize these signals more precisely within the hematopoietic hierarchy, we deployed a single-cell sequencing strategy (outlined in S2A Fig) which allowed us to map key subpopulation markers (S2B Fig) and used a "stem cell score" (see materials and methods and S2C Fig) to accurately define the response of primitive and committed cells to different concentrations of UM171 (Fig 1D). We next performed integrated gene expression analysis of primitive and committed cell subsets subjected to both low and high UM171 concentrations. As proof of principle, we observed that upregulation of HSC specific genes, most notably *PROCR* or *JAML [23]* was highest in the primitive compartment and followed a dose dependency to UM171 (Fig 1E). In addition, we noticed a marked increase in genes encoding MHC-associated proteins, most notably HLA-A and B, as well as beta 2-microglobulin (B2M) in all conditions compared to DMSO in all conditions compared to DMSO (S2D Fig). Since HLA upregulation is a hallmark response to pro-inflammatory signals, this suggested that UM171 induces an inflammatory stimulus in CB cells across several cell-types. Supporting this hypothesis, gene set enrichment analysis revealed pathways related to MHC protein complex binding, inflammation and NFkB signaling as dominant features of UM171 treatment (Fig 1F, S2E Fig and S1 Table).

Subpopulation analyses (primitive versus committed) showed common (e.g. HLA) and selective inflammatory responses. As shown in Fig 1G, inflammation genes such as BST2 and TNFSF10 appeared to be more induced in committed subpopulations upon UM171 exposure. In contrast, several members of the TNF/NFkB/cAMP signaling network were mostly induced by UM171 in the primitive subpopulation: These genes, known to be involved in fine-tuning the inflammatory response, included BIRC2-3, DUSP4, PDE4B, TIPARP (Fig 1H and S1 Table).

We previously reported the optimal UM171 dose-response for HSC expansion at around 35nM [19, 23]. Higher levels eventually lead to reduced cell proliferation and to elevated inflammation which is accompanied by a preferential reduction in cell proliferation in the primitive subset (see MKI67, GO cell cycle and CFSE labeling in S3A and S3B Fig). These "inflammation high / proliferation low" primitive cells failed to expand ex vivo (S3C and S3D Fig), potentially linking inflammation to reconstitution.

Given the observed specific inflammatory signature upon UM171 stimulation, we next assessed the possibility that this signature is secondary to induction (global profiling: S4A Fig and S1 Table) or secretion (secretion assay: S4B Fig) of pro-inflammatory factors such as TNFa or IFNResults shown in S4A and S4B Fig do not support this possibility.

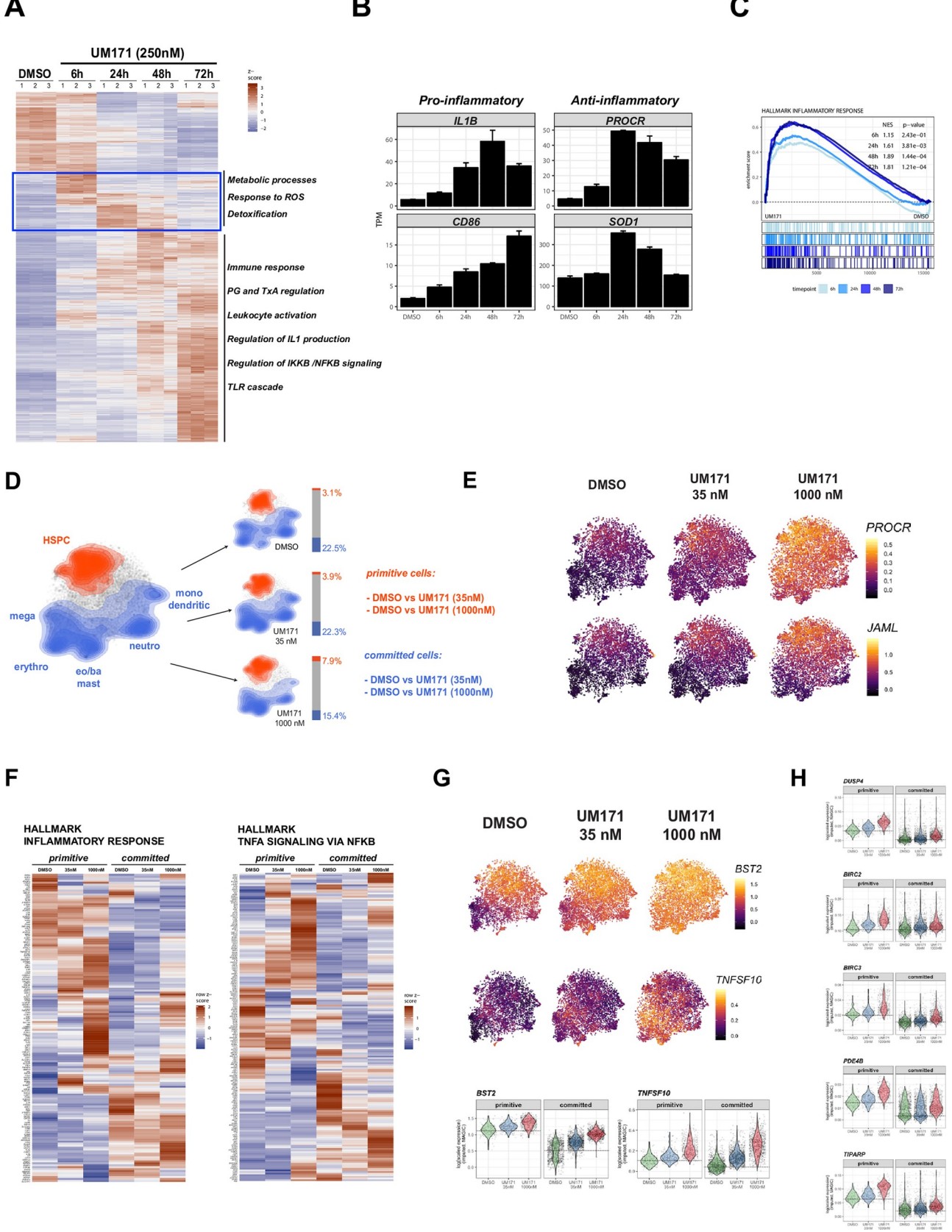

**Fig 1. UM171 exposure induces inflammatory gene signature. A:** Heat-map of differentially expressed transcripts after 6, 24, 48 and 72hrs of UM171 treatment in OCI-AML5 cell line (>2-fold change, q-value < 0.0001 at any timepoint compared to untreated). **B:** Examples of pro-inflammatory (IL1b and CD86) and anti-inflammatory genes (PROCR and SOD1) expression changes upon UM171 treatment in OCI-AML5 cells. **C:** GSEA plots showing enrichment for genes associated to inflammatory responses in OCI-AML5 cell line (250nM, 6, 24, 48 and 72hrs exposure).**D:** Topology of primitive (red) and committed (blue) cell subsets on top of t-SNE projection in either the combined dataset (left panel) or separated based on treatment (right panels). Barplots next to t-SNE graphs summarize cellular compositions of individual treatments into primitive/committed subsets. **E:** t-SNE heatmap of representative stem cell associated genes PROCR (also called EPCR or CD201) and JAML; imputed data (MAGIC). **F:** Heatmap summaries (average values) of selected inflammation associated genesets (corresponding GSEA in S2E Fig). **G:** t-SNE heatmap and corresponding violin plots (lower panel) of representative inflammatory genes modulated in response to UM171 in both primitive and committed cells (BST2 and TNFSF10); imputed data (MAGIC). **H:** Violin plots illustrating expression distribution of candidate genes associated to TNFa/NFkB pathway across primitive and committed subsets. Note the preferential UM171 mediated modulation of these candidate genes in primitive subset. p≤0.05.

Moreover, exposure of CD34$^+$ CB cells to low (10ng/ml) or high (50ng/ml) doses of TNFa or IFNg did neither cause upregulation of EPCR nor CD86 (S4C Fig), demonstrating the complexity of the UM171 inflammatory response which cannot be recapitulated by these 2 canonical pro-inflammatory agonists

## Proinflammatory signaling is essential for UM171 induced HSPC expansion

Dexamethasone (Dex) treatment was able to completely suppress the UM171 mediated expansion of HSPC-enriched cell subsets (CD34$^+$EPCR$^+$CD90$^+$CD45RA$^-$: see shNT panels in Fig 2A). This effect was specific to the glucocorticoid receptor as it was markedly attenuated in the absence of this receptor (see shGR panels in Fig 2A). Of interest, reduced HSPC expansion in response to Dex was also observed in the absence of UM171, indicating a general involvement of inflammation in HSCs (Fig 2B and 2C). Most importantly, Dex treatment also reduced the long-term repopulating ability of human HSCs and significantly abolished UM171-driven expansion of these cells (Fig 2D).

We also tested the combination of UM171 with the NFB inhibitor EVP4593 on CD34$^+$ cells. We observed a marked decrease in frequencies and numbers of CD34$^+$CD90$^+$EPCR$^+$ and CD34$^+$CD86$^+$ cells (Fig 2E and 2F), both in the presence or absence of UM171, suggesting that the expansion of these subsets depend on NFB signaling activation. Interestingly, NFkB inhibition significantly reduced EPCR gene expression in HSC enriched population (S5A Fig).

Consistent with these results, Dex as well as other anti-inflammatory inhibitors (JNK, NFkB and NFAT antagonists) suppressed UM171 mediated EPCR and CD86 induction in the OCI-AML5 cell line model (S5B–S5D Fig).

Since cyclic AMP acts as an important anti-inflammatory messenger by inhibiting NFkB signaling through PKA activation [25, 26], we also tested the impact of several cAMP elevating agents on UM171-mediated cord blood expansion. In line with above results, we found that cAMP elevation, whether using a phosphodiesterase inhibitor (IBMX), an adenylate cyclase activator (Forskolin) or cell permeable cAMP (db-AMP), dramatically reduced the proportion CD34$^+$EPCR$^+$ and CD34$^+$CD86$^+$ in UM171-supplemented cultures (S6 Fig).

Taken together, these results demonstrate that activation of pro-inflammatory programs, most notably of NFkB, represents a core requirement for UM171-driven *ex vivo* HSC expansion.

## EPCR attenuates UM171-mediated pro-inflammatory signals

We previously showed that EPCR expression is rapidly induced by UM171 in primitive CD34$^+$ CB cells and that this EPCR-positive subpopulation includes most short- and long-term HSCs. Moreover, we showed that EPCR is essential for the *in vivo* activity of human HSCs [23]. Since anti-inflammatory functions connected to NFkB activation have been

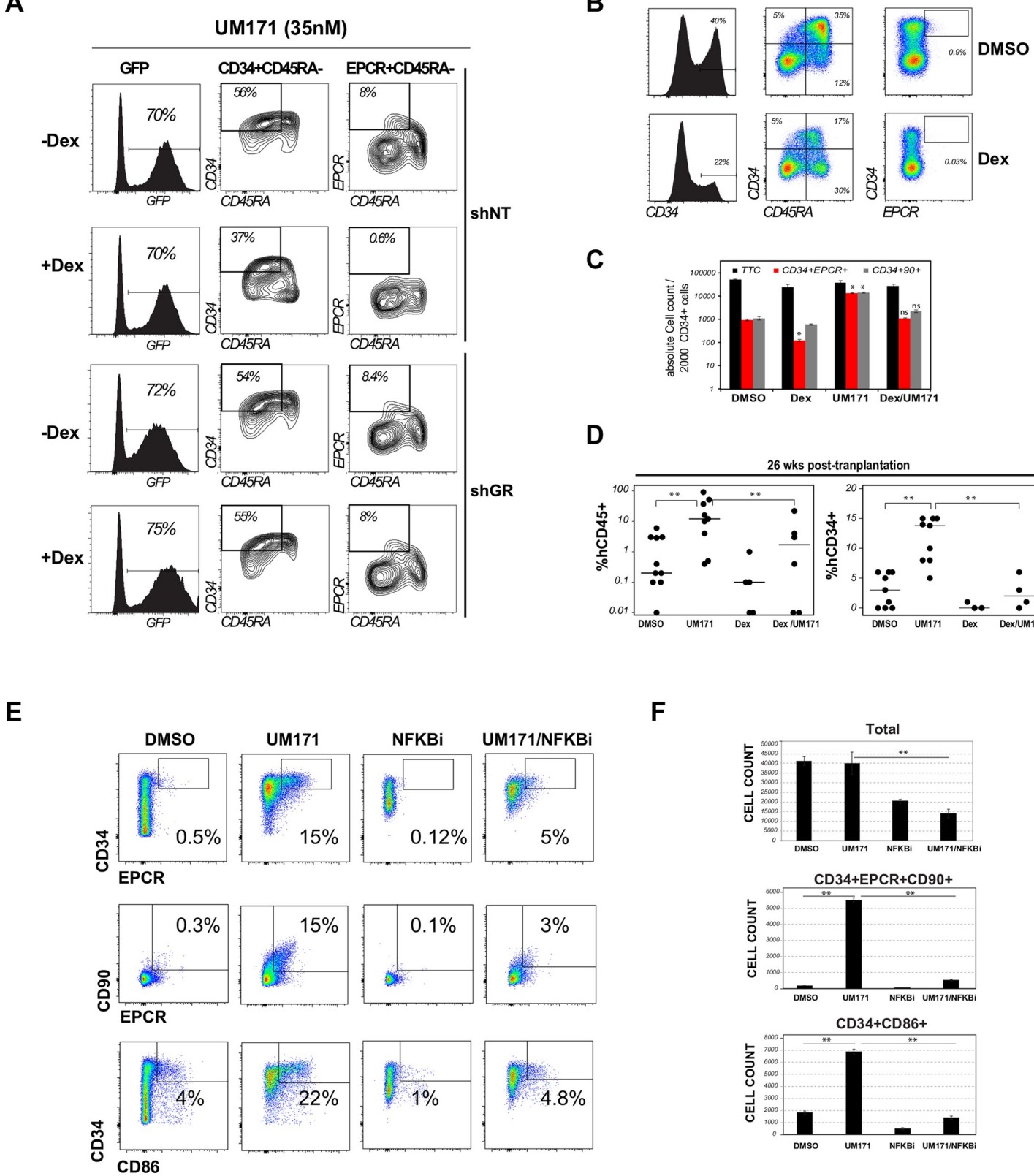

**Fig 2. Immunosuppressors abolish UM171 activity in HSPC. A:** CD34+ cord blood cells engineered to express shNT (GFP) or shNR3C1 (shGR) were cultured for 7 days with UM171 (35nM) in presence or absence of Dex (100nM). UM171-mediated expansion of HSC (CD34+CD45RA-EPCR+) subset from transduced (GFP+) cells

were evaluated by flow cytometry. **B:** CD34+ cord blood cells were cultured for 7 days in presence of vehicle or Dex only (100nM). Representative FACS profiles show a reduction of HSCs (CD34+EPCR+) and progenitors (CD34+ and CD34+CD45RA+) in presence of Dex. **C:** CD34+ cord blood cells were exposed to DMSO, UM171 (35nM), Dex (100nM) or UM171 and Dex for 7 days and co-stained with CD34, EPCR and CD90 antibodies. Total cell counts and counts of HSC enriched subsets are presented. Data show mean ±SEM of 2 independent experiments performed in triplicate. **D:** Day 7 cultures were transplanted in immunocompromised NSG mice (outcome of 2000 day 0 cells). Human CD45 and CD34 engraftment were assessed at 26 weeks post-transplantation. **E:** CD34+ cord blood cells were cultures for 5 days in presence of DMSO, UM171 (35nM), NFKB inhibitor (EVP4593, 100nM) and UM171 + EVP4593. FACS profile (left panel) and absolute counts (right panel) of CD34+EPCR+, CD34+EPCR+CD90+ and CD34+CD86+ cells are shown. Data show mean ±SEM of 2 independent experiments performed in triplicate.

reported for EPCR [27–29], we hypothesized that it might partially antagonize the inflammatory response in cells expanded under UM171 treatment. To test this hypothesis, we targeted the *EPCR* gene in the OCI-AML5 cell line using CRISPR (Fig 3A–3C). Strikingly, we observed a strong and proinflammatory/NFkB signature specifically in UM171 treated cells that was further exacerbated in three independent *sgEPCR* OCI-AML5 clones (Fig 3A–3C and S1 Table).

Of interest, EPCR knockout OCI-AML5 cells were hypersensitive to the combination of TNFa and UM171 (Fig 3D, left panel) which induced a hyper inflammatory response as detected by nitric oxide (NO) production (Fig 3D, right panel). In the context of primitive CD34+CD45RA- cord blood cells, UM171 and TNFa treatment also resulted in NO hyperproduction and cell loss when EPCR was experimentally reduced (Fig 3E). Together, these observations demonstrate that, in addition to causing a pro-inflammatory response, UM171 also triggers a critical negative feedback loop mediated by EPCR that protects HSPCs from inflammation borne cytotoxicity.

## UM171 activates a ROS detoxification program through EPCR

We noted that UM171 rapidly induced an EPCR-dependent ROS detoxification response in OCI-AML5 cells (blue rectangle in Figs 3A and 1A). Within 30 minutes of UM171 exposure (Fig 4A, top left panel), at a time where EPCR levels remain unaffected (Fig 4A, top right panel), ROS are rapidly induced and eventually regressed as EPCR levels rise (Fig 4A, bottom panels). At 24 hours, when detoxification genes become upregulated in an EPCR-dependent manner (Fig 4B and blue rectangle in Fig 3A), ROS levels fall below basal values in a UM171 dose dependent manner reaching their lowest at 125-250nM (bottom left panel, Fig 4A and S7A Fig).

Most importantly, UM171 also reduced ROS levels in primary CD34+ CB cells treated for 24h with various ROS inducing agents such as etomoxir, rotenone, oligomycine, CCCP (Fig 4C). At day 7, when expansion of functionally defined HSCs reaches its maximum [19, 23], steady-state levels of intracellular ROS were significantly blunted by UM171 in CD34+ CD45RA- cells in the presence of oligomycin (Fig 4D and 4E and S7B Fig) and under mild hyperthermia (S7C Fig). Moreover, a robust protective effect of UM171 treatment on ROS generation was observed most strikingly within the CD34+EPCR+ population (Fig 4F in blue, and Fig 4G left panel) versus the more mature CD34+CD45RA- subset (Fig 4D in red, and Fig 4E right panel). In summary, theses observations strongly indicate that UM171 protects against oxidative stress through EPCR induction.

## Discussion

In this study, we provide insights into how UM171 promotes the expansion of HSCs by coordinating pro- and anti-inflammatory responses. Interestingly, whereas pro-inflammatory cytokines like interferons or TNFa and anti-inflammatory drugs like dexamethasone are unable to establish permissive conditions for HSC expansion in culture, a low dose of UM171 (35nM) has the unique capacity to induce a rheostatic regulation of inflammatory and anti-inflammation/detoxification programs, hence enabling an effective context to promote HSC self-

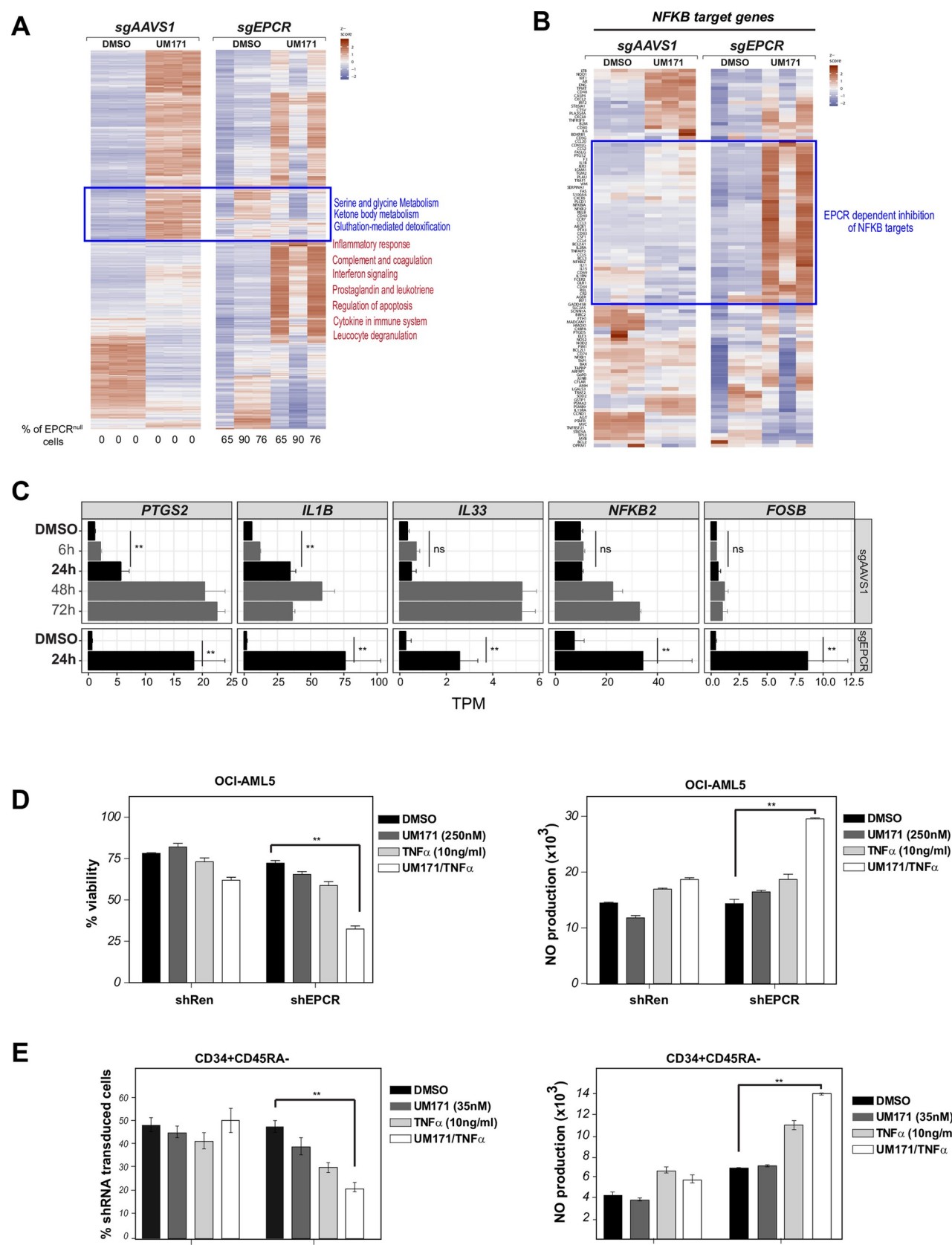

**Fig 3. EPCR attenuates UM171-mediated pro-inflammatory signals. A:** Heatmap representing differentially expressed genes after a 24 hours exposure to UM171 (250nM) in wild-type (sgAAVS1) and EPCR null (sgEPCR) context (>2-fold change, q-value < 0.0001 compared to DMSO/sgAAVS1). Note that while UM171 mediated pro-inflammatory signaling (red labels) is exacerbated in sgEPCR transduced cells, UM171 mediated-detoxification responses (blue labels) is hampered in absence of EPCR. **B:** Heatmap of NFkB target genes (http://bioinfo.lifl.fr/NF-KB) after a 24 hours exposure to UM171 (250nM) in wild-type (sgAAVS1) and EPCR null (sgEPCR) context. **C:** Examples of pro-inflammatory genes expression changes upon UM171 treatment (dashed lines show differential response to UM171 in EPCR null vs wild-type context after 24-hour exposure). **D:** OCI-AML5 were transduced with shRen (CT) or shEPCR (Ametrin) lentiviral vectors and culture for 4 days in presence of DMSO, UM171, TNF (10ng/ml) or UM171+ TNFa. Cell viability (left panel) and nitric oxide (NO) generation (right panel) were assessed by flow cytometry. **E:** CD34+ cord blood cells were transduced with shNT or shEPCR (Ametrin) lentiviral vectors and culture for 7 days in presence of DMSO, UM171, TNFa (10ng/ml) or UM171 + TNFa. Percentage of transduced Ametrin+ cells (right panel) and levels of nitric oxide (NO) (left panel) were assessed within the CD34+CD45RA- (HSPC enriched) population by flow cytometry. Data are expressed as mean ± SEM of results from 2 cord blood donors performed in 3 replicates.

renewal (Fig 4H). This inflammatory response is observed within 6 hours of UM171 treatment and does not appear secondary to pro-inflammatory cytokines such as TNFa. Our results thus suggest that UM171 establishes a dosage-dependent and tightly regulated inflammatory tonus that equally relies on positive regulators such as NFkB and their integration in a negative feed-back loop which prevents toxic accumulation of ROS/inflammation that is executed through upregulation of EPCR (Fig 4I). We also show that several of these components are active in cells not exposed to UM171, further extending these observations to HSC self-renewal networks. Future work will determine the mediator(s) of this specific inflammatory response.

Supporting a role for pro-inflammatory signaling in HSC self-renewal, exposure of cord blood cells to immunosuppressors such as glucocorticoids led to a marked reduction of HSC repopulating ability and abolished UM171-driven HSC expansion in a glucocorticoid receptor dependent manner. These observations contradict recent findings showing a beneficial effect of glucocorticoids on HSC engraftment [30]. It is important to note that duration of glucocorticoid exposure was different in the 2 studies. While a 16 hour exposure led to an enhanced HSC engraftment [30], longer treatment as performed in the present study had detrimental effects on HSC function. In line with this, while chronic low doses of the pro-inflammatory TLR ligand LPS dramatically impaired HSC function [31], short-term treatment with higher doses of LPS led to increased HSC multi-lineage repopulation ability [32]. It appears therefore that the effects of inflammation on HSCs activity are dose and time-dependent. Consistent with this, while transient NFkB blockade (6 to 24 hours) enhances *ex vivo* HSC propagation [33], a minimal and critical threshold of NFkB activation is required to maintain HSC homeostasis [34].

Interestingly, HSPCs and committed cells seemed to adapt a distinct inflammatory response upon UM171 treatment. Of interest, many of the differentially modulated genes in primitive cells (such as DUSP4 and BIRC2/3) are known to act as molecular brake in inflammation. It is thus tempting to speculate that HSPCs have established a fine-tuning inflammatory program to mount a robust protective response and avoid self-destructive inflammation. It will be critical in future works to dissect more precisely these inflammatory programs.

In summary, and supporting the work of several ongoing investigations, our current work supports a central role for inflammation in adult stem cell self-renewal, with UM171 providing a favorable state for blood stem cells renewal in which appropriate pro- and anti-inflammatory/detoxifying conditions are achieved.

## Materials and methods

### Human CD34+ cord blood cell collection

Umbilical cord blood units were collected from consenting mothers according to ethically approved protocol at Charles-Lemoyne Hospital, Montreal, QC, Canada. Human CD34+cord blood (CB) cells were isolated using The EasySep™ positive selection kit (StemCell

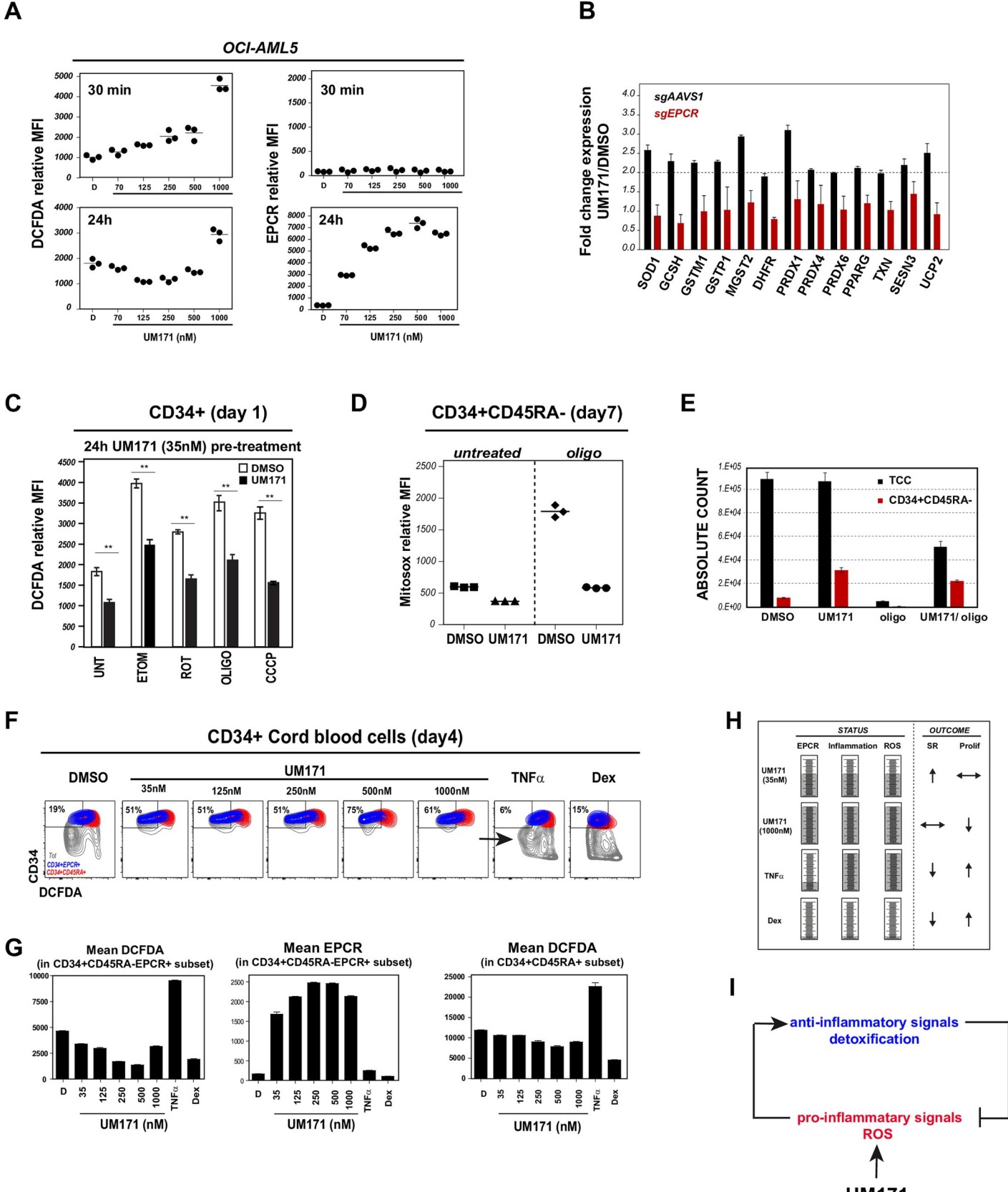

**Fig 4. UM171 activates a ROS detoxification program through EPCR. A:** Assessment of total ROS generation (left panels) in OCI-AML5 cells (as measured by DCFDA relative mean fluorescence intensity) and EPCR levels (right panels) at 30 minutes (upper panels) and 24h (lower panels) after addition of increasing UM171 concentration. **B:** Fold-change in expression of ROS detoxifying enzymes after exposure to UM171 in EPCR wild-type (sgAAVS1, black histograms) or EPCR null (sgEPCR, red histograms) context. **C:** CD34+ cord blood cells were exposed to DMSO or UM171 (35nM) for 24hrs and treated with different ROS-generating agents for 3 additional hours. ROS generation was then assessed by flow cytometry. **D:** CD34+ cord blood cells were exposed for 7 days to DMSO or UM171 (35nM) ± oligomycin. Mitochondrial ROS were then assessed in HSC enriched CD34+CD45RA- subset by flow cytometry using mitosox staining. Total and CD34+CD45RA- absolute cell counts are presented in **E. F:** CD34+ cord blood cells were cultured for 4 days in presence of DMSO or increasing dose of UM171, or pro-inflammatory cytokine TNFa (10ng/ml) or anti-inflammatory drug dexamethasone (100nM). ROS generation generated in each condition was evaluated using DCFDA staining. Representative FACS profile show ROS production in CD34+EPCR+ (HSC-enriched (blue)) vs CD34+CD45RA+ (progenitors (red)) subsets. **G:** Graphs showing relative ROS (left panel) and EPCR levels (middle) in HSC-enriched subset and ROS levels in committed cells (right panel). **H:** Summary of inflammatory status and effects on self-renewal (SR) and proliferation observed after exposure of different immunomodulatory drugs. **I:** Model depicting UM171-mediated activation of inflammation and subsequent induction of EPCR which in turn restricts excessive inflammation/ROS production.

Technologies Cat # 18056). Sorting for more primitive phenotypes was done in additional step using BD Aria II sorter.

## CD34+ cell culture

Human CD34+ cells were cultured in HSC expansion media consisting of StemSpan SFEM (StemCell Technologies) supplemented with human 100 ng/ml stem cell factor (SCF, R&D Systems), 100 ng/ml FMS-like trysine kinase 3 ligand (FLT3, R&D Systems), 50 ng/ml thrombopoietin (TPO, R&D Systems), and 10 microg/ml low-density lipoproteins (StemCell Technologies).

## Cell Lines

AML cell lines were purchased from the DSMZ German collection of Microorganisms and cell culture (Leibniz Institute) and the ATCC. OCI-AML3 and OCI-AML5 cells were cultured in MEM alpha media with 10% FBS and supplemented with 10 ng/mL of GM-CSF (PeproTech). NB4 and THP-1 cell lines were cultured in RPMI 1640 medium with 10% fetal bovine serum.

## Single cell RNA sequencing, population mapping and data analysis

CD34+ cord blood (CB) cells were cultured for 48h, a timepoint before most cells undergo division. Experimental conditions included either DMSO or two different UM171 concentrations; 35nM, a dose previously established as optimal for HSC expansion [19] as well as 1000 nM. Single-cell RNAseq from each of these cultures was performed on a Chromium Single-Cell Controller (10X Genomics) using the Single Cell 3' Reagent Kit version 2 according to manufacturer's instructions. Target cell numbers were 6,000 per condition. Sequencing of the scRNAseq libraries were performed on a NovaSeq device using a S2 (PE 28x91) setup. A standard Cellranger v3.0.1 pipeline was used for read mapping (GRCh38 annotation) and demultiplexing. Subsequent analyses were done in Seurat (v2.3) [35] and included (i) exclusion of cells with less than 1,000 genes or unique molecular identifiers (UMI), (ii) exclusion of cells with more genes or UMIs than the respective means plus 2 standard deviations (likely representing multiplets), (iii) exclusion of cells with more than 6% mitochondrial gene expression (representing apoptotic cells). Expression counts were log-normalized (scale factor 10,000) and scaled including regression on number of UMI's, cell cycle scores and mitochondrial gene content. Multi-set Canonical Correlation Clustering and tSNE embedding was performed using variable genes (defined by mean.function = ExpMean, dispersion.function = LogVMR, x.low. cutoff = 0.02, x.high.cutoff = 5, y.cutoff = 2) excluding sex-specific genes. Average fold-change and p-values were obtained using the FindMarkers function in Seurat with logFC threshold and min.pct set to zero. Log2 fold-change ranked gene lists were used for subsequent GSEA analyses. Significantly regulated genes were defined by p-values smaller than 0.01 and 1.5-fold

down or upregulation, respectively. For visualization purposes (S2B–S2D Fig and Fig 1E and 1G), data imputation was done using MAGIC (https://github.com/KrishnaswamyLab/MAGIC) with t = 1 [36].

Initial population mapping was performed based on expression of the stem and progenitor associated genes *AVP* and *HLF [23]* (see S2B Fig, HSPC panel), of genes indicative of early lymphoid fate (*SPINK2* and *SELL*; LMPP panel) and selective expression of myeloid, erythroid and megakaryocytic markers which were located to the periphery of the t-SNE projection (S2B Fig).

We next ranked cells using a stem-score (S2C Fig, top heatmap) that was calculated by averaging the normalized expression vectors of twelve commonly accepted stem cell specific genes [23, 37–39] (S2C Fig, middle heatmap). As expected, expression of differentiation associated genes anticorrelated with this ranking metric (S2C Fig, lower heatmap). Next, we categorized the top 5% stem-score ranking cells as 'primitive' and the lowest 20% as 'committed' subsets (Fig 1D, bottom). Notably, t-SNE representation of these designated populations recapitulated a clear spatial distinction between HSPCs in the top central (Fig 1D, in red) and committed/differentiated populations in the lower periphery of the projection space (Fig 1D, in blue). Albeit with small changes in frequency compared to the combined dataset, primitive and committed populations distributed well into all three experimental conditions (Fig 1D, middle panel).

## Flow cytometry analysis

Mouse anti-human antibodies were used to detect CD34 (APC or BV421- BD Biosciences), CD45RA (PE- BD Biosciences), CD86 (PerCP-eFluor710- eBioscience), CD90 (PECY7- BioLegend), and EPCR (APC-BioLegend). Flow cytometry acquisitions were performed on a Canto II cytometer (BD Biosciences) and data analysis was performed using FowJo software (Tree Star, Ashland, OR, USA) and GraphPad Prism software. Dead cells were excluded using 7AAD staining.

## Transplantation assays

All experiments with animals were conducted under protocols approved by the University of Montreal Animal Care Committee. EPCR cell subsets purified from uncultured or expanded CD34+CD45RA- CB cells were transplanted by tail vein injection into sub-lethally irradiated (250 cGy, <24 hr before transplantation) 8 to 16-week-old female NSG (NOD-Scid IL2Rgnull, Jackson Laboratory) mice. Human cells NSG-BM cells were collected by femoral aspiration or by flushing the two femurs, tibias and hips when animals were sacrificed at week 26.

## Nitric oxide production

Intracellular NO was measured after staining for cell surface markers, by incubation of cells for 15 min at 37˚C with 10 microM DAF-FM diacetate followed by extensive washes, according to the manufacturer's instructions (Molecular Probes).

## Cytokine assays

Cytokine levels in day 4 DMSO or UM171-exposed CD34+ cord blood cultures were measured using the LEGENDplex Human Inflammation Panel (13-plex) according to manufacturer's instructions. Data was analyzed using the LEGENDplexTM Data Analysis Software. In some conditions, cytokines secretion was also assessed after 4 hours PMA/ionomycin (20 ng/ml, 500 ng/ml) stimulation.

## Drugs

Dexamethasone (Dex), EVP (NFKB inhibitor), Forskolin, IBMX, db-cAMP, etomoxir, oligo-mycin, rotenone and CCCP were purchased from Sigma. SP600125 (JNK inhibitor) were purchased from selleckchem and cyclosporine A (CsA) were purchased from Cayman.

## shRNA and sgRNA vectors

shRNA vectors against EPCR and NR3C1 were reported previously [23, 40] respectively). 3 different sgRNA against EPCR were designed (sg1: ACAGCCCAGGAAGCAGCGGA; sg2: GGTGAAGGTGACCACTCCGG; sg3: GGGACACCTAACGCACGTGC).

## Bulk RNA sequencing and data analysis

3–5 x $10^5$ cells were FACS sorted from day 7 cultured cord blood derived CD34$^+$ HSPCs and preserved at -80 C in TRIzol Reagent (Thermo Fisher Scientific Cat # 15596026). cDNA libraries were constructed according to TruSeq Protocols (Illumina) and sequencing was performed using an Illumina HiSeq 2000 instrument. Gene expression statistics were obtained using the kallisto/sleuth analysis pipeline (https://pachterlab.github.io/sleuth/about) and the GRCh38 version 84 annotation. Differential gene expression was determined using sleuth p-values and fold-change in transcripts per million (TPM) values as designated. GSEA analysis was done using the fgsea R package and the full Molecular Signatures Database (MSigDB, Broad Institute).

## Accession codes

Gene Expression Omnibus: GSE57561, GSE138487 and GSE138680

## Supporting information

**S1 Fig. UM171 induces upregulation of EPCR and CD86 in leukemic cell lines. A:** Representative FACS profiles of CD86 and EPCR co-expression in monocytic derived cell lines exposed to DMSO or UM171 (500nM) for 24h. Various myeloid derived cell lines were screened for the upregulation of both EPCR and CD86 in response to UM171. Among them, acute myeloid leukemia cell lines OCI-AML3 and OCI-AML5 (FAB M4), promyelocytic leukemia NB4 (FAB M3) and monoblastic leukemia THP-1 (FAB M5) are shown. Note that OCI-AML5 cell line was used for all further studies as it shows the most consistent and highest response to UM171. **B:** Dose response curves for UM171-induced EPCR (upper panel) and CD86 (lower panel) expression in each cell lines (relative mean fluorescence intensity was assessed by flow cytometry). Data are expressed as mean ± SEM of 3 independent experiments.
(TIF)

**S2 Fig. UM171 exposure correlates with inflammation signature in CD34+ cells. A:** Experimental design to identify UM171 induced transcriptomic changes in single CD34+ cord blood cells. **B:** Combined t-SNE projections (grey dots) of a total of 16,669 CD34+ CB cells treated with either DMSO or two different doses of UM171 (35 and 1000 nM). Cell populations were identified by key marker expression and are plotted on top of t-SNE map. HSPC: hematopoietic stem and progenitor cells; LMPP: lymphoid primed multi-potent progenitors; mono/dendritic: mature monocytic/dendritic cells; neutro: neutrophils, eo/ba/mast: eosinophils/basophils/mast cells; erythro: erythoid cells; mega: megakaryocytic cells. Cellular phenotypes in the central t-SNE projection space exhibited less discrete but more transitionary gene

expression patterns (not shown), consistent with intermediate differentiation states and progressive lineage specification. **C:** Heatmap of stem cell associated genes across 16,669 cells used for calculation of a stem score, and selected differentiation genes. Bar plot (bottom) represents the cutoff for categorization into primitive and committed cell subsets. **D**: t-SNE heatmap of representative inflammatory genes B2M and HLA-A; imputed data (MAGIC). **E**: GSEA enrichment of selected inflammation associated genesets.
(TIF)

**S3 Fig. Impact of high dose UM171 exposure on HSPC. A:** GSEA enrichment summary indicating a selective cell cycle blockade in the primitive cell subset treated with 1000 nM UM171 (upper panel). Violin plots of distributions of expression levels of cell cycle gene MKI67 (lower panel). Note the selective reduction of MKI67-expressing cells in primitive UM171 (1000nM) treated subset (imputed single cell expression data). **B:** CD34+ cord blood cells were cultured for 4 days in presence of DMSO or UM171 (35nM and 1000nM). Percentage of CD34+ CD45RA- HSC enriched subset are shown in upper panel. Cell division of CD34+CD45RA- subsets was assessed using CFSE staining method (lower panel). Graph show % of cells in each generation. **C:** CD34+ cord blood cells were cultured for 7 days in presence of DMSO or UM171 (35nM and 1000nM). CD34+CD45RA- enriched HSC cell count were assessed before transplantation. **D:** Day 7 cultures exposed to DMSO or UM171 (35nM and 1000nM) were transplanted in immunocompromised NSG mice (outcome of 2 CRU). Human CD45 engraftment was assessed at 20 wks post-transplantation. Note that high dose of UM171 affect its capacity to expand HSCs with long-term repopulating activity.
(TIF)

**S4 Fig. UM171 inflammatory response is not recapitulated by pro-inflammatory agonists TNF and IFN. A:** Expression trajectories of interleukin, chemokine, interferon, TNF and TGFb family members in DMSO versus UM171 (35nM) treated CD34+ cord blood cells. Gene family annotations were downloaded from HUGO gene nomenclature committee (www.genenames.org). **B:** Amounts of pro-inflammatory cytokines IL1b, TNFa, IFNa2 and IFNg were measured by flow cytometry (LegendPlex) in day4 DMSO or UM171 exposed CD34+ culture media. Note that secretion of these pro-inflamatory cytokines were not induced by UM171 even after PMA/ionomycin stimulation. **C:** CD34+ cord blood cells were cultured for 4 days in presence of DMSO or UM171 (35 and 1000nM), or pro-inflammatory cytokine TNFa (10 and 50ng/ml) or IFNg (10 and 50ng/ml). CD34, EPCR and CD86 surface expression were assessed by flow cytometry. Representative FACS profile (upper panels) showing % of CD34+EPCR+ and CD34+CD86+ subsets and absolute counts (lower panels) of indicated populations in each condition.
(TIF)

**S5 Fig. Immunosuppressors abolish UM171 inflammatory response in leukemic cell lines. A:** Modulation of EPCR mRNA levels in response to NFKB inhibitor in enriched HSC subset. Data shown represent mean fold change in EPCR expression (± S.E.M.) of sorted CD34+ CD45RA- cells cultured for 48h in presence of DMSO, UM171 (35nM), NFKB inhibitor (EVP4593, 100nM) and UM171 + EVP4593 (representative of 2 independent specimen done in quadruplicates). **B:** Representative FACS profile (upper panel) and inhibition curves (lower panel) of UM171 mediated EPCR and CD86 induction after dexamethasone treatment. Data are shown as mean ± SEM for 2 independent experiments. **C:** GR knockdown was performed in OCI-AML5 cells. Transduced cells were exposed to DMSO, UM171 (250nM) or UM171 (250nM)/Dex (100nM) and EPCR and CD86 expression were evaluated by flow cytometry. Representative FACS profiles are shown in left panel and inhibition response to dexamethasone

are presented in right panel. Data are shown as mean ± SEM for 2 independent experiments. **D:** Percentage of UM171 induced CD86$^+$ EPCR$^+$ OCI-AML5 cells after 2 days treatment with following immunosuppressive drugs: (JNK inhibitor (SP600125, 200nM), NFKB inhibitor (EVP4593, 100nM), cyclosporine A (10nM).
(TIF)

**S6 Fig. cAMP elevating agents abolish UM171 inflammatory response in CD34+ cells. A:** CD34$^+$ cord blood cells were exposed to UM171 (35nM) in presence or absence of the indicated cAMP elevating agents (adenylate cyclase activator Forskolin (10microM), phosphodiesterase inhibitor IBMX (200microM) and cell permeable cAMP analog db-cAMP (100M)), cells were then assessed by flow cytometry for UM171 induced expression of expression of CD34, EPCR and CD86. Data show representative FACS profiles of 3 independent experiments.
(TIF)

**S7 Fig. UM171 prevents oligomycin and hyperthermia-induced ROS elevation in CD34+ cells. A:** OCI-AML5 cells stably expressing Dox-inducible MitoTimer vector were exposed to increasing doses of UM171 for 24hrs and treated with Dox for 3hrs. Cells were the analysed by flow cytometry. Data show representative dot-plot profile of MitoTimer expressing cells (y-axis, green channel; x-axis, red channel). Note that while low dose of UM171 (125 to 500nM) reduces signals in red channel (consistent with lower ROS and improved mitochondrial quality), high dose of UM171 (above 1microM) increase ROS level (enhanced red signal). **B:** Representative FACS profile of ROS production at day 7 in CD34+ cells exposed or not to UM171 ± oligomycin. **C:** CD34+ cord blood cells were exposed for 7 days to DMSO or UM171 (35nM) in mild hyperthermia condition. Mitochondrial ROS were then assessed in HSC enriched CD34+CD45RA- subset by flow cytometry using mitosox staining.
(TIF)

**S1 Table. AML5 and CD34+ Transcriptome and single cell RNAseq data.**
(XLSX)

## Acknowledgments

The authors would like to thank M. Frechette and V. Blouin-Chagnon for assistance with in vivo experiments, I. Boivin for CD34$^+$ cord blood cells purification, G. Dulude and A. Gosselin at the Institute of Research in Immunology and Cancer for technical support with flow cytometry sorts. The authors also thank Charles-Le Moyne Hospital for providing human umbilical cord blood units. This work was supported by grants from the Canadian Institutes of Health Research and the Stem Cell Network of Canada (G.S.). E. Tomellini and L. Simon were supported by a fellowship from the Cole Foundation. J.F.S was supported by IVADO scholarship.

## Author Contributions

**Conceptualization:** Jalila Chagraoui, Guy Sauvageau.

**Data curation:** Jean Francois Spinella.

**Formal analysis:** Jalila Chagraoui, Bernhard Lehnertz, Simon Girard, Jean Francois Spinella, Iman Fares, Elisa Tomellini.

**Funding acquisition:** Guy Sauvageau.

**Investigation:** Jalila Chagraoui, Simon Girard, Iman Fares, Elisa Tomellini.

**Methodology:** Jalila Chagraoui, Simon Girard, Jean Francois Spinella, Nadine Mayotte, Tara MacRae.

**Project administration:** Guy Sauvageau.

**Software:** Bernhard Lehnertz, Jean Francois Spinella.

**Supervision:** Jalila Chagraoui.

**Validation:** Jalila Chagraoui, Simon Girard, Nadine Mayotte, Sophie Corneau, Tara MacRae, Laura Simon.

**Writing – original draft:** Jalila Chagraoui.

**Writing – review & editing:** Bernhard Lehnertz, Jean Francois Spinella, Guy Sauvageau.

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
