## [Decision Letter · Decision Letter 0]

25 Aug 2019

PONE-D-19-19512

UM171 induces a homeostatic inflammatory-detoxification response supporting human HSC self-renewal

PLOS ONE

Dear Dr. Sauvageau,

Thank you for submitting your manuscript to PLOS ONE. After careful consideration, we feel that it has merit but does not fully meet PLOS ONE’s publication criteria as it currently stands. Therefore, we invite you to submit a revised version of the manuscript that addresses the points raised during the review process.

We would appreciate receiving your revised manuscript by 45 days. To enhance the reproducibility of your results, we recommend that if applicable you deposit your laboratory protocols in protocols.io, where a protocol can be assigned its own identifier (DOI) such that it can be cited independently in the future. For instructions see: http://journals.plos.org/plosone/s/submission-guidelines#loc-laboratory-protocols

We look forward to receiving your revised manuscript.

Kind regards,

Maria Cristina Vinci, PharmD, PhD

Academic Editor

PLOS ONE

Journal Requirements:

2. To comply with PLOS ONE submissions requirements, in your Methods section, please provide additional information on the animal research and ensure you have included details on (1) methods of sacrifice, (2) methods of anesthesia and/or analgesia, and (3) efforts to alleviate suffering.

3. Our internal editors have looked over your manuscript and determined that it may be within the scope of our 'Stem Cell Plasticity in Tissue Repair and Regeneration' Call for Papers. This collection of papers is headed by a team of Guest Editors for PLOS ONE, and we hope to bring together researchers working on a wide range of disciplines, from molecular, cellular and developmental topics to preclinical and clinical work. Additional information can be found on our announcement page: https://collections.plos.org/s/stem-cell . If you would like your manuscript to be considered for this collection, please let us know in your cover letter and we will ensure that your paper is treated as if you were responding to this call. Agreeing to be part of the call-for-papers will not affect the date your manuscript is published. If you would prefer to remove your manuscript from collection consideration, please specify this in the cover letter.

4. We note that you are reporting an analysis of a microarray, next-generation sequencing, or deep sequencing data set. PLOS requires that authors comply with field-specific standards for preparation, recording, and deposition of data in repositories appropriate to their field. Please upload these data to a stable, public repository (such as ArrayExpress, Gene Expression Omnibus (GEO), DNA Data Bank of Japan (DDBJ), NCBI GenBank, NCBI Sequence Read Archive, or EMBL Nucleotide Sequence Database (ENA)). In your revised cover letter, please provide the relevant accession numbers that may be used to access these data. For a full list of recommended repositories, see http://journals.plos.org/plosone/s/data-availability#loc-omics or http://journals.plos.org/plosone/s/data-availability#loc-sequencing.

Reviewers' comments:

Reviewer's Responses to Questions

**Comments to the Author**

1. Is the manuscript technically sound, and do the data support the conclusions?

Reviewer #1: Yes

2. Has the statistical analysis been performed appropriately and rigorously? 

Reviewer #1: Yes

3. Have the authors made all data underlying the findings in their manuscript fully available?

Reviewer #1: Yes

4. Is the manuscript presented in an intelligible fashion and written in standard English?

Reviewer #1: Yes

5. Review Comments to the Author

Reviewer #1: Chagraoui et al investigated the roles of UM171 in HSCs and found that UM171 triggers a balanced pro- and anti-inflammatory network that relies on NFKB activation and EPCR-dependent detoxification. This represents a new finding with mechanistic insights. Major comments:

1. Fig 1: the authors may explain why they chose to use OCI-AML5 cell line. What was the culture condition for this cell line - is it different from that for culture of HSCs? Do medium components affect UM171’s effect on OCI-AML5 cells, which enables the gene expression profiles very different from those of UM171 treated hCB CD34+ cells? In addition, please include the description of cell type used in Fig 1d-h in the figure legends.

2. Does NFkB inhibitor decrease EPCR mRNA in HSCs?

3. Given that committed cells respond to UM171 differently from primitive cells, the authors may comment on whether UM171 has different effects on hCB CD34+ cells and purified hCB HSCs in terms of stem cell expansion.

6. PLOS authors have the option to publish the peer review history of their article (what does this mean?). If published, this will include your full peer review and any attached files.

Reviewer #1: Yes: Chengcheng Zhang

---

## [Author Response · Author response to Decision Letter 0]

16 Oct 2019

Review Comments to the Author and Responses

Reviewer #1: Chagraoui et al investigated the roles of UM171 in HSCs and found that UM171 triggers a balanced pro- and anti-inflammatory network that relies on NFKB activation and EPCR-dependent detoxification. This represents a new finding with mechanistic insights. Major comments:

We thank reviewer #1 for his insights and have addressed each of the points raised.

1. Fig 1: The authors may explain why they chose to use OCI-AML5 cell line. 

Various myeloid derived cell lines were screened for the upregulation of both EPCR and CD86 in response to UM171 (considered as a signature of UM171 activity in primary hematopoietic cells). Among them, Acute Myeloid Leukemia cell lines OCI-AML3 and OCI-AML5 (FAB M4), promyelocytic leukemia NB4 (FAB M3) and monoblastic leukemia THP-1 (FAB M5) are shown in supplementary Figure 1. OCI-AML5 cell line was chosen for further studies (transcriptome analysis, ROS production, sensitivity to inflammation…) as it shows a high clonogenecity and has the most consistent and highest response to UM171 (based on EPCR and CD86 surface expression).

All informations concerning cell lines are now in Supplementary Figure 1 legend and Materials and Methods sections.

What was the culture condition for this cell line - is it different from that for culture of HSCs? 

For all experiments performed in this study, OCI-AML5 cells were cultured in presence of 10% serum and GM-CSF (10ng/ml) according to manufacturer’s. CD34+ cord blood cells were cultured in serum free medium supplemented with SCF, TPO and FLT3L cytokines as we have previously shown that this condition is necessary to achieve optimal HSC expansion and that UM171 cannot compensate for any of these cytokines (Fares I and al, 2014). Unfortunately, culture of OCI-AML5 in serum free medium (as performed for cord blood cells) dramatically affects its survival independently of UM171 presence.

Do medium components affect UM171’s effect on OCI-AML5 cells, which enables the gene expression profiles very different from those of UM171 treated hCB CD34+ cells?

Our work shows that UM171 exhibit inflammatory properties in both OCI-AML5 cells and primitive CD34+ cells and any inflammatory insult will affect UM171 effects. As shown in Figure 3, pro-inflammatory cytokines such as TNF��� exacerbate UM171 effect on both OCI-AML5 and CD34+ cells especially in the absence of EPCR anti-inflammatory properties.

In addition, please include the description of cell type used in Fig 1d-h in the figure legends.

Done

2. Does NFkB inhibitor decrease EPCR mRNA in HSCs?

Yes. NFKB inhibitor (EVP4593) significantly decreased EPCR mRNA levels in HSC enriched CD34+CD45RA- subset. See new supplementary Figure 5A and text.

3. Given that committed cells respond to UM171 differently from primitive cells, the authors may comment on whether UM171 has different effects on hCB CD34+ cells and purified hCB HSCs in terms of stem cell expansion.

This question was addressed in our previous paper “EPCR expression marks UM171-expanded CD34+ cord blood stem cells. Fares I et al, Blood. 2017 Jun 22; 129(25):3344-3351. doi: 10.1182/blood-2016-11-750729. Epub 2017 Apr 13. PMID: 28408459. ” (See main Figure 2).

In this previous study, we have shown that total day3 UM171-expanded cord blood cells and day3 UM171-expanded EPCR+ sorted cells (highly enriched in HSC) exhibit similar in vitro (Figure 2C) and in vivo (Figure 2D) expansion, suggesting that in this setup the effect of UM171 on committed cells (CD34+EPCR-) have little if no impact on HSC subset expansion. 

Importantly, using UM171 in combination with a “fed-batch” culture strategy (Csaszar E et al, Cell Stem Cell. 2012 Feb 3; 10(2):218-29. doi: 10.1016/j.stem.2012.01.003. PMID: 22305571) allows the dilution of inhibitory factors produced by committed cells and hence optimal HSC expansion.

---

## [Decision Letter · Decision Letter 1]

24 Oct 2019

UM171 induces a homeostatic inflammatory-detoxification response supporting human HSC self-renewal

PONE-D-19-19512R1

Dear Dr. Sauvageau,

We are pleased to inform you that your manuscript has been judged scientifically suitable for publication and will be formally accepted for publication once it complies with all outstanding technical requirements.

With kind regards,

Maria Cristina Vinci, PharmD, PhD

Academic Editor

PLOS ONE

Additional Editor Comments (optional):

Reviewers' comments:

Reviewer's Responses to Questions

**Comments to the Author**

1. If the authors have adequately addressed your comments raised in a previous round of review and you feel that this manuscript is now acceptable for publication, you may indicate that here to bypass the “Comments to the Author” section, enter your conflict of interest statement in the “Confidential to Editor” section, and submit your "Accept" recommendation.

Reviewer #1: All comments have been addressed

2. Is the manuscript technically sound, and do the data support the conclusions?

Reviewer #1: Yes

3. Has the statistical analysis been performed appropriately and rigorously? 

Reviewer #1: Yes

4. Have the authors made all data underlying the findings in their manuscript fully available?

Reviewer #1: Yes

5. Is the manuscript presented in an intelligible fashion and written in standard English?

Reviewer #1: Yes

6. Review Comments to the Author

Reviewer #1: The authors have satisfactorily addressed all the reviewer's previous comments. The paper has been strengthened.

7. PLOS authors have the option to publish the peer review history of their article (what does this mean?). If published, this will include your full peer review and any attached files.

Reviewer #1: Yes: Cheng Cheng Zhang

---

## [Editor Report · Acceptance letter]

31 Oct 2019

PONE-D-19-19512R1 

UM171 induces a homeostatic inflammatory-detoxification response supporting human HSC self-renewal 

Dear Dr. Sauvageau:

I am pleased to inform you that your manuscript has been deemed suitable for publication in PLOS ONE. Congratulations! Your manuscript is now with our production department. 

With kind regards,

on behalf of

Dr Maria Cristina Vinci 

Academic Editor

PLOS ONE